# Dual Transcriptome Analysis Reveals the Changes in Gene Expression in Both Cotton and *Verticillium dahliae* During the Infection Process

**DOI:** 10.3390/jof10110773

**Published:** 2024-11-07

**Authors:** Yongtai Li, Yuanjing Li, Qingwen Yang, Shenglong Song, Yong Zhang, Xinyu Zhang, Jie Sun, Feng Liu, Yanjun Li

**Affiliations:** The Key Laboratory of Oasis Eco-Agriculture, Agriculture College, Shihezi University, Shihezi 832003, China; liyongtai@stu.shzu.edu.cn (Y.L.); liyuanjing@stu.shzu.edu.cn (Y.L.); 20232112028@stu.shzu.edu.cn (Q.Y.); songshenglong@stu.shzu.edu.cn (S.S.); 20222012025@stu.shzu.edu.cn (Y.Z.); zhxy@shzu.edu.cn (X.Z.); sunjie@shzu.edu.cn (J.S.)

**Keywords:** cotton, *Verticillium dahliae*, dual RNA-sequencing, pathogen–host interaction

## Abstract

Cotton is often threatened by Verticillium wilt caused by *V. dahliae*. Understanding the molecular mechanism of *V. dahlia*–cotton interaction is important for the prevention of this disease. To analyze the transcriptome profiles in *V. dahliae* and cotton simultaneously, the strongly pathogenic strain Vd592 was inoculated into cotton, and the infected cotton roots at 36 h and 3 d post infection were subjected to dual RNA-seq analysis. For the *V. dahliae*, transcriptomic analysis identified 317 differentially expressed genes (DEGs) encoding classical secreted proteins, which were up-regulated at least at one time point during infection. The 317 DEGs included 126 carbohydrate-active enzyme (CAZyme) and 108 small cysteine-rich protein genes. A pectinesterase gene (VDAG_01782) belonging to CAZyme, designated as *VdPE1*, was selected for functional validation. *VdPE1* silencing by HIGS (host-induced gene silencing) resulted in reduced disease symptoms and the increased resistance of cotton to *V. dahliae*. For the cotton, transcriptomic analysis found that many DEGs involved in well-known disease resistance pathways (flavonoid biosynthesis, plant hormone signaling, and plant–pathogen interaction) as well as PTI (pattern-triggered immunity) and ETI (effector-triggered immunity) processes were significantly down-regulated in infected cotton roots. The dual RNA-seq data thus potentially connected the genes encoding secreted proteins to the pathogenicity of *V. dahliae*, and the genes were involved in some disease resistance pathways and PTI and ETI processes for the susceptibility of cotton to *V. dahliae*. These findings are helpful in the further characterization of candidate genes and breeding resistant cotton varieties via genetic engineering.

## 1. Introduction

Cotton, a primary source of natural fibers, is extensively cultivated worldwide [1,2,3]. China is the world’s major cotton producer and largest cotton consumer. The cotton industry plays an important role in the development of China’s national economy and is the main source of income for farmers in cotton-producing areas. However, cotton production often suffers from the harm of diseases. Verticillium wilt, caused mainly by *Verticillium dahliae*, poses a significant challenge to cotton cultivation and production. Microsclerotia, a dormant structure of *V. dahliae*, can persist in soil for more than 14 years and serve as the initial source of infection [4,5]. When soil conditions are favorable, microsclerotia can germinate and produce infectious mycelium, which invades the plant’s internal tissues from the root epidermis and colonizes the vascular tissues, causing the characteristic wilting symptoms of *V. dahliae* infection, such as leaf yellowing, wilting, abscission, etc. [6,7]. Verticillium wilt is difficult to control due to the stable microsclerotia structure and diverse physiological races of *V. dahliae*. In recent years, with the completion of sequencing of *V. dahliae* and cotton, great progress has been made in the exploration and functional characterization of pathogenicity-related genes in *V. dahliae* [8] and disease resistance-related genes in cotton [9]. Exploring the interaction mechanism between host and pathogen has become a research hotspot [10].

Pathogens secrete a wide range of biomolecules such as proteins, toxins, and secondary compounds during infection [11]. The secreted proteins are synthesized by pathogens in cells, then transported to the cell surface and plasma membrane space through various secretion pathways to sense the surrounding environment and make various adaptive responses [12]. The secreted proteins contain a large number of carbohydrate-active enzymes (CAZymes), including glycoside hydrolases (GHs), glycosyltransferases (GTs), polysaccharide lyases (PLs), carbohydrate esterases (CEs), and carbohydrate-binding modules (CBMs) [13]. Cell wall-degrading enzymes (CWDEs), belonging to GHs, are particularly important for the pathogenic process of pathogens [14]. Previous studies have shown that several CWDE genes, such as *VdPEL1*, *Vdxyn4*, and *VdCUT11*, serve as virulence factors in the pathogenic process of *V. dahliae* [15,16,17]. Additionally, some secreted proteins have been found to act as PAMP (pathogen-associated molecular pattern) molecules or effectors to stimulate or inhibit plant immune responses by interacting with proteins in plants. For instance, *VdPEL1*, *Vdxyn3*, and *VdCUT11* can also serve as PAMPs to activate plant immune responses [15,16,17]. The small secreted protein VdSCP7 is an effector that can trigger cotton immune responses [18]. The effector proteins VdSCP27, VdSCP113, and VdSCP126 can stimulate tobacco immunity, leading to cell necrosis and the up-regulation of resistance-related gene expression [19]. The CFEM-class effectors, VdSCP76 and VdSCP77, can suppress the host’s immune response, facilitating pathogen infection [20].

During the long-term co-evolution process between plants and pathogenic microorganisms, plant cells initiate two types of immune responses, pattern-induced immunity (PTI) and effector-induced immunity (ETI) [21,22]. Plants directly recognize the pathogen-associated molecular patterns (PAMPs) of pathogens through pattern recognition receptors (PRRs) located on the cell membrane, triggering the plant’s first layer of PTI to resist pathogen infection [23]. Plant PRRs include receptor-like kinases (RLKs) and receptor-like proteins (RLPs) [24]. Successful invading pathogens can secrete effectors into plant cells, thereby attacking the plant’s immune system for further invasion. Plants directly or indirectly recognize the effectors of the pathogen through the resistance proteins inside the cell, triggering the plant’s second layer of the immune system, ETI, activating a stronger immune response to resist pathogen invasion [25]. Resistance proteins can be classified into five types based on the conservation of their gene structure, including nucleotide-binding leucine-rich repeat (NB-LRR) proteins, leucine-rich repeat (LRR) proteins, serine/threonine protein kinases (Ser/Thr Kinase, STK), leucine-rich repeat protein kinases (LRR-STK), and disease resistance proteins with only a coiled-coil domain (CC) [26]. Currently, several PRRs and resistance proteins have been reported to contribute to the resistance of cotton to *V. dahliae*, such as *GhMPK9*, *GhRVD1*, and *GhDSC1* [9,27,28].

RNA-seq has been widely used to study plant–pathogen interactions in many crops [29,30,31], including interactions between cotton and *V. dahliae* [32,33,34]. It is notable that most of the studies mentioned above were limited to a single transcription study of a plant or pathogen. For cotton–*V. dahliae* interaction studies, most of the studies focused on transcriptome analysis of cotton during infection, as it was difficult to collect *V. dahliae* samples from cotton roots. Recently, dual RNA-seq technology, which can simultaneously sequence and analyze the transcriptome profiles in both plants and pathogens, has proven to be a powerful tool to study in vivo interactions between pathogen and their host plants. Dual RNA-seq technology has been used in some crops [35,36,37], and it is necessary for the study of the interaction between cotton and *V. dahliae*. In this study, we used a dual RNA-seq analysis to investigate the changes in gene expression in *V. dahliae* and cotton at two time points (36 h and 3 d post infection) to discover the potential pathogenic factors in *V. dahliae* and the defense response of cotton during infection, providing a basis for us to better understand the molecular mechanism of interaction between cotton and *V. dahliae*.

## 2. Materials and Methods

### 2.1. Fungal Strains, Plant Material, and Culture Conditions

The wild-type *Verticillium dahliae* strain Vd592 was initially cultured on potato dextrose agar (PDA) medium for 7 days. The strain was then inoculated into complete medium (CM) liquid medium and incubated at 25 °C with a shaking speed of 200 rpm for 7 days. For long-term storage, the fungal suspension was mixed with glycerol at a ratio of 2:3 (strain to glycerol) and preserved at −80 °C. A susceptible upland cotton variety, Xinluzao 7, was used in this study. It was planted in an incubator at 28 °C under a photoperiod of 16 h light/8 h dark and a relative humidity at 60%. The model plant *N. benthamiana* was grown in a greenhouse at 25 °C under a photoperiod of 16 h light/8 h dark and a relative humidity at 60%.

### 2.2. Dual RNA-Sequencing (RNA-Seq)

The fungus was cultured in CM liquid medium at 25 °C with shaking at 200 rpm for 7 days. The conidial suspension was filtered through four layers of gauze and adjusted to 1 × 10^7^ CFU/mL with sterilized water using a hemacytometer. When the susceptible cotton variety Xinluzao 7 reached the two-true-leaf stage, the conidial suspension (1 × 10^7^ CFU/mL) was inoculated to the cotton roots using the root-irrigation method as previously reported [38]. The cotton plants were treated with water as a control. Cotton roots were harvested at 36 h and 3 d post inoculation, respectively. The total RNA was extracted using an RNA prep Pure Plant Kit (Tiangen, Beijing, China), and RNA-seq libraries were generated using a Hieff NGS Ultima Dual-mode mRNA Library Prep Kit for Illumina Sample Preparation (Yeasen, Shanghai, China). Dual RNA-sequencing was performed on the Illumina NovaSeq platform at Biomarker Technology Co. (Beijing, China). The trimming and quality control of raw reads were performed using the BMKCloud bioinformatics analysis platform (www.biocloud.net, accessed on 26 September 2024). The clean data were aligned with the *Verticillium dahliae* reference genome (VdLs.17) and the cotton reference genome (TM-1) using the Hisat2 tool version 2.0.4. StringTie version 2.2.1 was employed for transcript assembly based on the reference annotation (RABT), identifying known transcripts and predicting novel ones from the Hisat2 alignment results.

### 2.3. Differentially Expressed Gene (DEG) Screening and Analysis

Differentially expressed genes were screened using edgeR software version 3.32.1 with the criteria at *p*-value ≤ 0.05 and |log2FC| > 1. Gene ontology (GO) enrichment analyses and Kyoto Encyclopedia of Genes and Genomes (KEGG) pathway enrichment for differentially expressed genes were performed using the online tools available at the Microbiotics website (https://www.bioinformatics.com.cn/, accessed on 9 September 2024). Secretory proteins were predicted based on SignalP-5.0, TMHMM 2.0, and SecretomeP 2.0 [39,40,41]. PHI (pathogen–host interaction) homologues were predicted using the PHI-base database [42]. The annotation of putative carbohydrate-active enzymes (CAZymes) was performed using a Hidden Markov Model (HMM) approach based on the Carbohydrate-Active EnZymes database [43].

### 2.4. Gene Expression Validation by Quantitative Real-Time PCR (qRT-PCR)

The total RNA was extracted using an RNA prep Pure Plant Kit (Tiangen, Beijing, China) and transcribed into cDNA using TransScript^®^ One-Step gDNA Removal and cDNA Synthesis SuperMix (TransGen Biotech, Beijing, China). The specific primers for each gene are listed in Appendix A. *V. dahliae* β-tubulin (VDAG_10074) and *GhUBQ7* (DQ116441.1) cotton were used as the internal reference genes. The qRT-PCR assays were performed with PerfectStart Green qPCR SuperMix (TransGen Biotech, Beijing, China) on a LightCycler 480II instrument (Roche, Indianapolis, IN, USA). The qRT-PCR reaction system consisted of 2 µL of template cDNA (100 ng), 0.4 µL each Forward and Reverse Primer (10 µM), 10 µL of ChamQ Universal SYBR qPCR Master Mix (2×), and 7.2 µL of Nuclease-free Water. The amplification procedure was as follows: initial denaturation at 95 °C for 30 s, followed by 40 cycles at 95 °C for 10 s and at 60 °C for 30 s. The relative expression levels of the genes were calculated using the 2^−ΔΔCt^ method.

### 2.5. HIGS (Host-Induced Gene Silencing) Treatment

The pTRV1, pTRV2, and pTRV2-*GhCHLI* plasmids were kindly provided by Prof. Longfu Zhu of Huazhong Agricultural University. The total RNA of *V. dahliae* Vd592 was extracted using a Fungal RNA Kit (Omega Inc., Guangzhou, China) and transcribed into cDNA using TransScript^®^ One-Step gDNA Removal and cDNA Synthesis SuperMix. The *VdPE1* (VDAG_01782) interference fragment (304 bp) was amplified from Vd592 cDNA and cloned into the pTRV2 vector. The resulting pTRV2-*VdPE1* vector was then transformed into the *Agrobacterium tumefaciens* strain GV3101 by electroporation and used for TRV (Tobacco Rattle Virus) treatment according to previous description (Appendix A) [44]. After *A. tumefaciens* carrying pTRV1 and pTRV2-*VdPE1* were simultaneously injected into cotton cotyledons, RNA1 and RNA2 of TRV could be generated from the two plasmids, respectively. RNA1 and RNA2 recognized each other to form dsRNA (double-strand RNA), which was then cleaved into siRNA (small interfering RNA) by dicer-like enzymes. Then, the siRNA could bind with RNA-induced silencing complex (RISC) and guide RISC to localization to mRNA strands complementary to siRNA, thereby achieving a gene silencing effect. The pTRV2-G*hCHLI* (the gene mutation causes the leaves to lose their green color) plasmid was used as a control.

When the TRV-treated cotton seedlings reached the two-true-leaf stage, they were infected with conidial suspension (1 × 10^7^ CFU/mL) via the root-irrigation method [38]. Disease symptoms, vascular bundle browning, and the cotton disease index were investigated at 14 and 21 dpi (days post inoculation). Infected plants were scored on a scale of 0 to 4 as previously reported [44]. The disease index (DI) was calculated using the following formula: DI = [(Σ disease grade × number of infected plants)/(total number of sampled plants × 4)] × 100. For fungal biomass detection and the gene silencing efficiency test, the roots of cotton plants treated with pTRV2-00 and pTRV2-*VdPE1* were collected separately at 21 dpi and used for total DNA extraction using the CTAB method and RNA extraction using an EASYspinPlus Plant RNA Extraction Kit (Aidlab, Beijing, China). The fungal biomass was quantified by qRT-PCR using the cotton roots’ DNA as template, *V. dahliae* β-tubulin (VDAG_10074) and cotton *GhUBQ7* (DQ116441.1) as internal reference genes, and ITS1-F/ST-Ve1-R as primers (Appendix A). The expression level of *VdPE1* was analyzed by qRT-PCR using cotton roots’ cDNA as a template, *V. dahliae* β-tubulin (VDAG_10074) as an internal reference gene, and *VdPE1*-qF/qR as primers (Appendix A).

### 2.6. Yeast Signal Sequence Trap System

To assess the function of the signal peptide (SP), the yeast signal sequence trap system was used. Briefly, the predicted *VdPE1* signal peptide coding region was cloned into the pSUC2 vector using specific primers (Appendix A) to generate pSUC2-*VdPE1*^sp^. This construct was then transformed into the *Saccharomyces cerevisiae* strain YTK12, which lacks the sucrose invertase gene. The *VdPE1* coding region without a signal peptide sequence was also cloned into the pSUC2 vector to generate pSUC2-*VdPE1*^Δsp^, serving as a negative control. The positive control was pSUC2-Avr1b^SP^. Transformed yeast strains were selected on CMD-W (tryptophan-deficient) medium (SS/-Trp with agar). Positive colonies were cultured on YPRAA medium containing 2% marshmallow. The secretory function of the SP was evaluated on YPRAA medium containing 2% marshmallow. Additionally, 2,3,5-triphenyltetrazolium chloride (TTC) was reduced to red insoluble 1,3,5-triphenylformazan (TPF), which was used to assay the transforming enzyme activity of the signal peptide. Transformed yeast strains were cultured in CMD-W medium for 24 h at 30 °C; they were then harvested by centrifugation and suspended with 1mL sterile water. After centrifugation, the fungal cells were resuspended with 1 mL of 10% sucrose solution and supplemented with 1mL of 2% TTC. After being placed in a water bath (30 °C) for 30 min and left at room temperature for 5 min, the color change was observed visually and photographed. Invertase activity was determined by the color change in TTC.

### 2.7. Agrobacterium Infiltration Assays

The full-length coding sequence of *VdPE1*, including the signal peptide sequence, was inserted into the PYBA-1132 vector and transformed into *Agrobacterium* strain GV3101 using heat shock transformation (Appendix A). For transient expression in *N. benthamiana*, GV3101 carrying the target vector PYBA-1132:*VdPE1* was resuspended in MES buffer (10 mM MgCl_2_, 10 mM MES, 10 μM acetosyringone), and the OD600 nm was adjusted to 0.6–0.8. The suspended *A. tumefaciens* cells were injected into 3-week-old *N. benthamiana* leaves. The negative control was PYBA-1132, and the positive control was PYBA-1132:BAX. Cell death was observed and photographed 7 days after injection. For cell death suppression analysis, the 3-week-old *N. benthamiana* leaves were injected with a 1:1 mixture of PYBA-1132:BAX and PYBA-1132:*VdPE1* after being kept in a dark environment for 3 h.

### 2.8. Statistical Analyses

Three biological replicates were performed for each treatment of experiments. Statistical analyses were conducted using SPSS statistical software version 26.0 (IBM, Armonk, NY, USA). A one-way analysis of variance (ANOVA) followed by the Student–Newman–Keuls (SNK) test with a *p*-value of 0.05 was used to determine significant differences between treatments.

## 3. Results

### 3.1. Gene Expression Changes in Vd592 and Cotton During Infection

To investigate the changes in gene expression both in *V. dahliae* and cotton during infection, dual transcriptome analyses were performed on cotton root systems infected with wild-type strain Vd592 at 36 h and 3 d post infection, and clean data were aligned with either the *V. dahliae* genome (VdLs.17) or the cotton genome (TM-1). PCA analysis revealed that the data from the three biological replicates were tightly clustered, indicating that the repeated experiments yielded similar results (Figure 1a,e), and thereby validating the repeatability of the experiments. Moreover, the PCA analysis indicated significant differences in the distribution of Vd592 samples before (0 hpi) and after infection (36 hpi and 3 dpi) and in the distribution of infected and water-treated cotton samples (Figure 1a,e), suggesting that infection status and time significantly impact gene expression both in Vd592 and cotton.

For Vd592 samples, a total of 4537 DEGs (2263 up-regulated and 2274 down-regulated) and 4693 DEGs (2376 up-regulated and 2317 down-regulated) were identified in Vd-36h vs. 0 and Vd-3d vs. 0 comparisons, respectively (Figure 1b). For the up-regulated DEGs, 525 were differentially expressed only at 36 hpi and 638 were only at 3 dpi, while 1738 were at both time points (Figure 1c). For the down-regulated DEGs, 536 were differentially expressed only at 36 hpi and 579 were only at 3 dpi, while 1738 were at both time points (Figure 1d). For Vd592-infected cotton root samples, a total of 19,374 DEGs (7756 up-regulated and 11,618 down-regulated) and 20,423 DEGs (8331 up-regulated and 12,092 down-regulated) were identified in X7-36h-T (infected cotton) vs. M (water treatment) and X7-3d-T vs. M comparisons, respectively (Figure 1f). For the up-regulated DEGs, 2139 were differentially expressed only at 36 hpi and 2714 were only at 3 dpi, while 5617 were at both time points (Figure 1g). For the down-regulated DEGs, 3264 were differentially expressed only at 36 hpi and 3738 were only at 3 dpi, while 8354 were at both time points (Figure 1h). The RNA-seq results were confirmed to be reliable by qRT-PCR using 12 randomly selected cotton and *V. dahliae* genes (Appendix A, Appendix A).

### 3.2. GO Enrichment and KEGG Pathway Analysis of DEGs Identified in V. dahlia and Infected Cotton Roots

To investigate the function of up- and down-regulated DEGs identified in Vd592 and infected cotton root samples, we performed gene ontology (GO) and Kyoto Encyclopedia of Genes and Genomes (KEGG) analyses. The top 10 GO terms and KEGG pathways are shown in Figure 2. For the up-regulated DEGs in Vd592, the most significant change in the cellular component category was observed in the integral component of the membrane term (GO:0016021), followed by the extracellular region term (GO:0005576). In the molecular function category, more DEGs were enriched in the transmembrane transporter activity term (GO:0022857). In the biological process category, most DEGs were associated with the transcription and DNA-templated terms (GO:0006351) (Figure 2a). KEGG analysis for the up-regulated DEGs showed that a high proportion of DEGs were involved in valine, leucine, and isoleucine degradation (ko00280); arginine and proline metabolism (ko00330); beta-alanine metabolism (ko00410); tryptophan metabolism (ko00380); starch and sucrose metabolism (ko00500); ascorbate and aldose metabolism (ko00053); and pentose and glucuronide interconversions (ko00040) pathways (Figure 2b). For the down-regulated DEGs in Vd592, most DEGs were associated with ribosomes (GO:0005840) in the cellular component category, most DEGs were involved in the structural components of ribosomes (GO:0003735) in the molecular functions category, and the majority of DEGs were associated with ribosomal small-subunit biogenesis (GO:0042274) in biological processes (Figure 2c). KEGG analysis revealed that the down-regulated pathways mainly included ribosomes (ko03010), alanine, aspartate and glutamate metabolism (ko00250), the biosynthesis of amino acids (ko01230), the pentose phosphate pathway (ko00030), carbon metabolism (ko01200), and amino sugar and nucleotide sugar metabolism (ko00520) (Figure 2d).

For the up-regulated DEGs in infected cotton roots, the most significant term was the cytoplasm (GO:0005737) in the cellular component category, followed by the obsolete cytoplasmic part term (GO:0044444). In terms of molecular function, more DEGs were enriched in FAD binding (GO:0071949), followed by protein serine/threonine phosphatase activity (GO:0004722). In the bioprocess category, most DEGs were associated with glutamine metabolism processes (GO:0006541) (Figure 2e). KEGG analyses showed that these up-regulated DEGs were mainly involved in valine, leucine, and isoleucine degradation (ko00280), fatty acid degradation (ko00071), arginine and proline metabolism (ko00330), β-Alanine metabolism (ko00410), and the peroxisome (ko04146) pathways (Figure 2f). For the down-regulated DEGs in infected cotton roots, DEGs were found to be mainly enriched in plasma membrane (GO:0005886) and microtubule (GO:0005874) terms in the cellular component category. In the molecular function category, microtubule binding (GO:0008017) and microtubule motor activity (GO:0003777) terms were significantly enriched. In the biological process category, most DEGs were enriched in the auxin-activated signaling pathway (GO:0009734) and microtubule-based movement (GO:0007018) terms (Figure 2g). KEGG analysis revealed that the down-regulated pathways mainly included plant hormone signal transduction (ko04075), the biosynthesis of amino acids (ko01230), carotenoid biosynthesis (ko00906), starch and sucrose metabolism (ko00500), the ABC transporter (ko02010), phenylpropanoid biosynthesis (ko00940), and flavonoid biosynthesis (ko00941) (Figure 2h).

### 3.3. Pathogenicity-Related Genes in the Vd592 Transcriptome

All the DEGs were annotated to identify the genes potentially related to the pathogenicity of *V. dahliae*, including secreted proteins (SPs), small cysteine-rich proteins (SCRPs) (<400 amino acids, ≥4 cysteine residues), carbohydrate-activating enzymes (CAZymes) and pathogen–host interactions (PHIs). A total of 1650 secreted proteins were identified, containing 556 classical secreted proteins (CSPs) and 1094 non-classical secreted proteins (N-CSPs). Of the 556 CSPs, 187 and 195 were CAZymes and SCRPs, respectively (Figure 3a). A hierarchical clustering heat map of the 1650 secreted protein genes showed that several SP genes were down-regulated, while more SP genes were up-regulated during infection (Figure 3b). Compared to 0hpi, a total of 455 differentially expressed CSP genes (238 up-regulated and 217 down-regulated) were identified at 36 hpi, and 490 differentially expressed CSP genes (280 up-regulated and 210 down-regulated) were identified at 3 dpi (Figure 3c,d).

The up-regulated genes during infection were often considered to be required for the pathogenicity of *V. dahliae*. A total of 317 up-regulated CSP genes were identified from 556 CSP genes, with 37 up-regulated only at 36 hpi, 79 at 3 dpi, and 201 at both time points (Figure 4a). Among the 317 up-regulated CSP genes, 126 were annotated to be carbohydrate-active enzyme genes, including 11 up-regulated only at 36 hpi, 43 at 3 dpi, and 72 at both time points (Figure 4b). The hierarchical clustering heat map showed that 126 CAZyme genes were significantly up-regulated at 36 hpi, 3 dpi, or both (Figure 4c). These up-regulated CAZymes included 67 glycoside hydrolases (GHs), 22 auxiliary activities (AAs), 18 polysaccharide lyases (PLs), 18 carbohydrate esterases (CEs) and 1 carbohydrate-binding module (CMB) (Figure 4d). Further classification found that there were several cell wall-degrading enzyme genes among the 126 CAZyme genes, including 15 cellulase, 21 pectinesterase, and 8 hemicellulase genes (Figure 4e), most of which (28) showed more than five-fold changes in expression level at least at one time point during infection, indicating their importance for the pathogenic process of *V. dahliae*.

Among the 317 up-regulated CSP genes, there were 108 SCRPs, including 12 up-regulated only at 36 hpi, 31 at 3 dpi, and 65 at both time points (Figure 5a). The hierarchical clustering heat map showed that 108 SCRP genes were significantly up-regulated at 36 hpi, 3 dpi, or both (Figure 5b). The 108 SCRP genes mainly included 4 CFEM structural domain proteins, 2 Ecp2 effector protein domain-containing proteins, 3 Carboxylic ester hydrolases, 2 GPI anchored proteins, 3 Metalloproteasse, 2 Mannan endo-l,4-beta-mannosidases, and several CWDE genes, such as 10 pectate lyases, 4 cutinases, 4 glucanases, 2 xylanases genes, etc. (Figure 5c). In addition, the remaining 69 SCRP genes have not been reported yet and need further research (Appendix A). Among the 144 PHI genes, only 32 were found to be up-regulated during infection, with 7 up-regulated at 36 hpi, 5 at 3 hpi, and 20 at both time points (Figure 5d). The hierarchical clustering heat map showed that 32 HPI genes were significantly up-regulated at 36 hpi, 3 dpi, or both (Figure 5e). Of the 32 up-regulated PHI genes, 13 were reported in *Verticillium dahliae*, and 19 were highly overlapping in homologous gene sequences in other reported fungi (Table 1).

### 3.4. Functional Verification of the Cell Wall-Degrading Enzyme Gene VdPE1 by HIGS

Based on transcriptome data analysis, a DEG (VDAG_01782) encoding cell wall-degrading enzyme, designated as VdPE1, was selected for functional verification using TRV-based host-induced gene silencing (HIGS). When the TRV-treated seedlings reached the two-leaf stage, they were infected with Vd592 conidial suspension (1.0 × 10^7^ CFU/mL), and disease symptoms were investigated at 14 and 21 dpi. Compared with the pTRV2-00 plants, the pTRV2-VdPE1-treated seedlings showed milder disease symptoms, including lighter necrosis, wilting, and vascular discoloration (Figure 6a), as well as a lower disease index (Figure 6b). Moreover, pTRV2-VdPE1 treatment showed a significant reduction in fungal biomass in the roots, stems, and leaves of the seedlings (Figure 6c). The qRT-PCR analysis showed that the expression level of the VdPE1 gene in pTRV2-VdPE1-treated plants was significantly lower than that in the pTRV2-00-treated plants at 14 dpi, indicating successful silencing of the VdPE1 gene in *V. dahliae* (Figure 6d).

### 3.5. VdPE1 Is a Secreted Protein and Fails to Induce Cell Death in Nicotiana benthamiana

VdPE1 encodes a 420 amino acid protein containing an N-terminal signal peptide that is 20 residues long (amino acid 1-20), suggesting that VdPE1 may be a secretory protein (Figure 7a). The VdPE1 signal peptide region was cloned into the pSUC2 vector to generate pSUC2-VdPE1^SP^. Additionally, the remaining region of VdPE1 was cloned into the pSUC2 vector to generate pSUC2-VdPE1^ΔSP^, which was used as a negative control. pSUC2-Av1rb^SP^ was used as a positive control. The pSUC2-VdPE1^SP^ and pSUC2-VdPE1^ΔSP^ vectors were individually transformed into the yeast strain YTK12. The results showed that only YTK12 carrying pSUC2-VdPE1^SP^ grew well on YPRAA media with raffinose as the sole carbon source. The insoluble 2,3,5-triphenyltetrazolium chloride (TTC) was reduced to insoluble red 1,3,5-triphenylcarboxylic acid hydrazide (TPF) after adding YTK12 carrying pSUC2-VdPE1^SP^, suggesting that VdPE1 is most likely to be secreted into the extracellular region under certain conditions (Figure 7b).

To analyze whether VdPE1 can elicit plant cell death, we cloned the CDS sequence containing the signal peptide into the vector pYBA-1132 to generate pYBA-1132:VdPE1, and then performed transient expression in *N. benthamiana* using an Agrobacterium-mediated transformation (agroinfiltration) technique. The pYBA-1132:GFP and PYBA-1132:BAX (Bcl-2-associated X protein) vectors were used as negative and positive controls, respectively. Agrobacterium carrying each vector was injected into the leaves of 3-week-old *N. benthamiana*. Seven days after injection, it was found that VdPE1 failed to elicit cell death or inhibit BAX-induced cell death (Figure 7c).

### 3.6. DEGs Associated with Flavonoid Biosynthesis in Cotton Transcriptome

KEGG analysis found that the flavonoid biosynthesis pathway was significantly down-regulated in infected cotton. A total of 84 DEGs were involved in flavonoid biosynthesis, mainly including four C4Hs (cinnamate 4-hydroxylases), seven CHSs (chalcone synthases), five CHI (chalcone isomerases), four F3Hs (flavanone-3-hydroxylases), two (flavonoid-3-plus monooxygenases), three ANSs (anthocyanin synthases), two ANRs (anthocyanin reductases), four DFRs (dihydroflavonol-4-reductases), and four LARs (colorless anthocyanin reductases), as shown in Figure 8. The heatmap showed that the expression of some key genes involved in the flavonoid biosynthesis pathway were significantly down-regulated at 36 hpi, 3 dpi, or both. The expression of most genes showed more than 5-fold down-regulation during infection, and some even showed more than 10-fold down-regulation, such as two CHIs (Gohir.D09G215050 and Gohir.D04G012300), two CHSs (Gohir.D02G031600 and Gohir.D10G144300), etc. (Figure 8). Flavonoid biosynthesis pathways have been found to play an important role in plant disease resistance [75]. Therefore, the down-regulation of this pathway may be responsible for the severe disease symptoms of cotton after infection.

### 3.7. DEGs Associated with Plant Hormone Signal Transduction in Cotton Transcriptome

KEGG pathway analysis also found that the plant–pathogen interaction pathway and plant hormone signal transduction pathway were down-regulated in infected cotton. For the plant–pathogen interaction pathway, a total of 545 genes were identified, including 58 involved in pathogen-associated molecular pattern-triggered immunity (PTI) and 5 involved in effector-triggered immunity (ETI). Genes related to PTI mainly included 5 RLKs (repeat receptor-like kinases), 2 LRR2 (leucine-rich repeat 2) genes, 9 RLP12 (receptor-like protein 12) genes, 5 BAK1 (BRI1-Associated Kinase 1) genes, 11 LecRKs (Lectin receptor-like kinases), 5 LRK10 (receptor-like kinase 10) genes, 4 LYM1 (Lysin Motif-containing protein 1) genes, 3 LYKs (LysM receptor-like kinases), 1 CNGC4 (Cyclic Nucleotide-Gated Ion Channel 4) gene, 5 CPKs (Calcium-dependent protein kinases), 2 FLS2 (flagellin sensing 2) genes, 3 MAPKs (mitogen-activated protein kinases), 1 CCAMK (Calcium and Calmodulin-Dependent Protein Kinase), 1 XA21 (Xanthomonas resistance 21) gene, and 1 PBL21 (PBS1-Like 21) gene. Genes related to ETI included five RPS5 (resistance to pseudomonas syringae 5) genes. The heatmap indicated that most of these genes were significantly down-regulated at both 36 hpi and 3 dpi (Figure 9a).

For the plant hormone signal transduction pathway, a total of 576 genes were identified, including IAA (indole-3-acetic acid), ETH (ethylene), CTK (cytokinin), GA (gibberellin), SA (salicylic acid), JA (jasmonic acid), ABA (abscisic acid), and brassinosteroid (BR) signal transduction pathways. Of these pathways, the SA and JA signaling pathways are widely considered to play important roles in plant disease resistance [76]. Therefore, the DEGs associated with the SA and JA signaling transduction pathways were focused on. A total of 40 DEGs were identified in the JA signaling pathway, including 8 JAR1 (jasmonate-responsive gene 1), 1 COI1 (coronatine insensitive 1), 10 JAZ (jasmonate ZIM-domain), and 21 MYC2 transcription factor genes. Additionally, 18 DEGs were involved in the SA pathway, including 3 NPR1 (non-expressor of pathogenesis-related 1) genes, 4 PR-1 (pathogenesis-related 1) genes, and 11 TGA (TGACG motif-binding factor) genes. The heatmap indicated that most of these genes were significantly down-regulated at both 36 hpi and 3 dpi (Figure 9b).

## 4. Discussion

Host infection by a pathogen involves two interacting organisms, the host and the pathogen, with significant differences in their transcription levels. Transcriptome sequencing typically examines the changes in the stress response of the plant and pathogen independently. RNA-seq studies on *V. dahliae* have been widely reported. However, we found that the *V. dahliae* samples used in RNA-seq were usually cultured by media [36], plant tissues [77], or root exudates [78] rather than isolated from plants. Using *V. dahliae* samples cultured by different media as materials, transcriptome sequencing has revealed changes in gene expression during the growth and development of this pathogen, providing a basis for understanding the pathogenic molecular mechanism of *V. dahliae*. However, these transcriptome sequencing results cannot reflect the changes in gene expression in *V. dahliae* during infection. To fully comprehend the interaction mechanism between cotton and *V. dahliae*, it is essential to analyze the transcriptional profiles in both the plant and pathogen simultaneously. Dual RNA-seq technology, or interaction transcriptome sequencing, offers a novel perspective for this purpose. This technology can isolate the transcripts of the host and pathogen in one step, which is not only simple and fast, but also significantly reduces the error rate of previous physical separations [79]. In this study, we used a dual RNA-seq analysis to investigate the changes in gene expression in both *V. dahliae* and cotton at two time points (36h and 3d post infection) to discover the potential pathogenic factors in *V. dahliae* and resistant factors in cotton during infection, providing a basis for us to better understand the molecular mechanism of interaction between cotton and *V. dahlia*.

With the completion of genome sequencing of *V. dahliae* and the development of bioinformatics tools, there were increasing reports on genes involved in the growth, development, and pathogenicity of *V. dahliae*. *VdPKS*, *VdT4HR*, *VdT3HR*, *VdGARP1*, and *Vayg1* genes were found to be involved in the microsclerotia formation and pathogenic process of *V. dahliae* [80,81,82,83]. *VdEXG* and *VdSec22* are related to cell wall degradation [84,85]. *VdSCP41*, *VdpevD1*, *Vd424Y*, and *VdNLP2* encode effector proteins which were found to be involved in the pathogenic process [49,86,87,88]. *VdFTF1*, *VdAtf1*, and *VdMRTF1* encode transcriptional factors regulating pathogenic genes [34,50]. However, the genes required for *V. dahliae* infection still need to be identified due to the complexity of the pathogenic molecular mechanism of the pathogen. Secreted proteins play a crucial role in the complex interactions between plants and pathogenic fungi, with classical secreted proteins being particularly significant in this dynamic ‘tug-of-war’ [89]. In this study, we found that many genes encoding secreted proteins were up-regulated during infection, suggesting that they were required for the infection process. These genes included 126 CAZYmes and 108 SCRPs. Some genes with the same name as DEGs have been reported to be involved in the pathogenesis of this pathogen. For instance, endopolygalacturonases *VdEG1* and *VdEG3* function as PAMPs (pathogen-associated molecular patterns) to activate the plant immune response and also serve as virulence factors that diminish the disease resistance of cotton [90]. The effector proteins *VdCE11* and *VdCE51* are pivotal in inducing and suppressing cell necrosis, respectively [91].

When cotton is subjected to pathogen infection, it rapidly activates a series of defense responses to resist pathogen invasion. The well-known defense pathways mainly included flavonoid biosynthesis, plant hormone signaling, and plant–pathogen interactions pathways [92]. Many disease resistance-related genes have been identified from cotton. *GhDFR1* involved in flavonoid biosynthesis was found to be crucial for cotton resistance to *V. dahliae* [93]. *GhPAL6*, *GhCOMT1*, *Gh4CL30*, and *GhCAD35* involved in lignin biosynthesis help resist the pathogen attack by reinforcing the cotton cell wall [94,95,96]. *GhERF* and *GhJAZ2* are involved in the plant hormone signaling pathway, which regulate the adaptability and responsiveness of cotton to environmental changes through a complex network [97,98]. A PRR gene, Calcium ion protein kinase *GhCPK33*, was found to be vital for cotton resistance to *V. dahliae* [99]. Members of the mitogen-activated protein kinase (MPK) family, *GhNTF6* and *GhMKK2*, are central to the detection of pathogen attack and the subsequent relay of defense signals in cotton [100,101]. Disease resistance protein R genes, including *GhDSC1*, *GbaNA1*, and *GbCNL130*, were found to directly implicated in the recognition and defense of cotton against pathogens [28,102,103]. This study found that after infection with the highly pathogenic strain Vd592, the flavonoid biosynthetic pathway, plant hormone signaling pathway, and plant–pathogen interaction pathway in cotton were significantly down-regulated, which was responsible for the obvious disease symptoms of susceptible variety Xinluzao 7. Several genes involved in these pathways showed obvious down-regulation in cotton, including several reported genes, such as CHS, LRR, RLP, and PR genes, providing candidates for molecular breeding for the disease resistance of cotton to *V. dahliae*.

## 5. Conclusions

This study applied dual RNA-sequencing to explore gene expression in both cotton and *Verticillium dahliae* at 36 h and 3 d post infection. For *V. dahliae*, 317 DEGs encoding classical secreted proteins were found to be up-regulated during the infection process, including 126 carbohydrate-active enzymes (CAZymes) and 108 small cysteine-rich protein genes (SCRPs), which may be responsible for the pathogenicity of *V. dahliae*. Of these DEGs, one encoding pectinesterase (*VdPE1*) was selected and proven to be associated with the pathogenicity of *V. dahliae* by using HIGS (host-induced gene silencing). For cotton, many DEGs involved in well-known disease resistance pathways and PTI (pattern-triggered) and ETI (effector-triggered immunity) processes were significantly down-regulated in infected roots. The results advance our understanding of the molecular mechanism of cotton–*V. dahliae* interaction and provide candidates for breeding resistant cotton varieties via genetic engineering.

## Figures and Tables

**Figure 1 jof-10-00773-f001:**
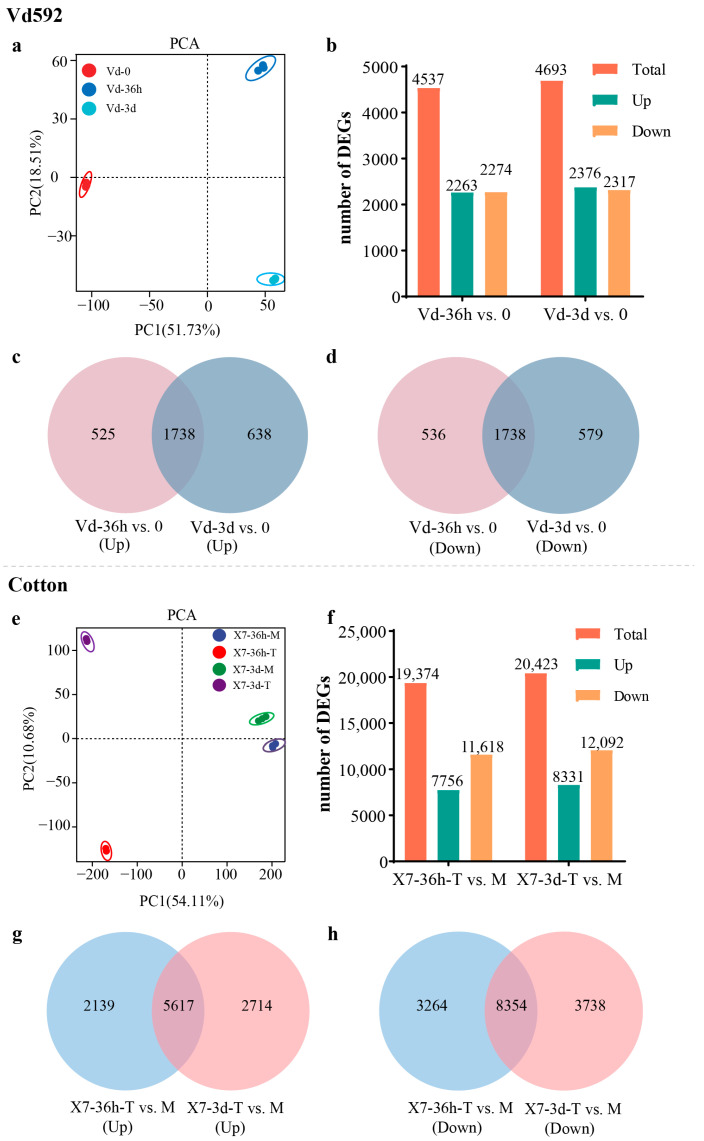
Overview of the transcriptome sequencing in *V. dahliae* and cotton at 36 h and 3 d post infection. (**a**) PCA analysis of *V. dahliae* samples. (**b**) Number of DEGs in Vd-36h vs. 0 and Vd-36d vs. 0 comparisons. (**c**) Venn diagram of up-regulated DEGs in Vd-36h vs. 0 and Vd-3d vs. 0 comparisons. (**d**) Venn diagram of down-regulated DEGs in Vd-36h vs. 0 and Vd-3d vs. 0 comparisons. (**e**) PCA analysis of cotton root samples. (**f**) Number of DEGs in X7-36h-T vs. M and X7-3d-T vs. M comparisons. (**g**) Venn diagram of up-regulated DEGs in X7-36h-T vs. M and X7-3d-T vs. M comparisons. (**h**) Venn diagram of down-regulated DEGs in X7-36h-T vs. M and X7-3d-T vs. M comparisons. In X7-36h-T vs. M and X7-3d-T vs. M comparisons, ‘T’ represents infected cotton, and ‘M’ represents Mock (water treatment).

**Figure 2 jof-10-00773-f002:**
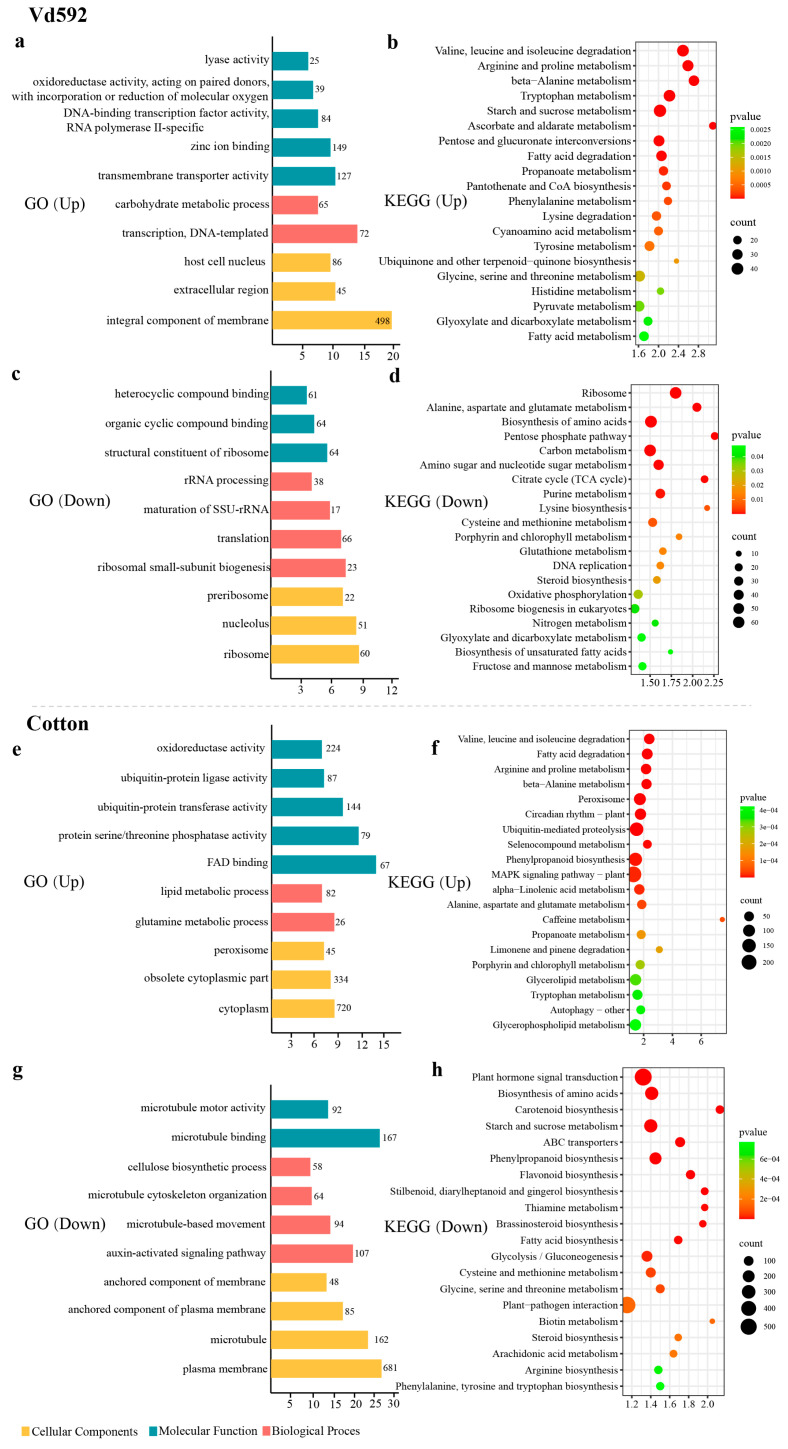
GO and KEGG enrichment analysis of differentially expressed genes (DEGs) in both *V. dahliae* and cotton roots. (**a**,**e**) GO enrichment analysis of up-regulated DEGs in *V. dahliae* and cotton roots. The x-axis represents the −log10 (*p* value), and the y-axis represents the top 10 enriched GO terms. The number next to each horizontal bar indicates the number of enriched DEGs within the corresponding GO term. (**b**,**f**) KEGG pathway analysis of up-regulated DEGs in *V. dahliae* and cotton roots. The x-axis represents the enrichment factor, and the y-axis lists the top 20 pathways. The color of each bullet indicates the *p*-value, and the size of the bullet reflects the number of enriched DEGs within the corresponding pathway. (**c**,**g**) GO enrichment analysis of down-regulated DEGs in *V. dahliae* and cotton roots. (**d**,**h**) KEGG pathway analysis of down-regulated DEGs in *V. dahliae* and cotton roots.

**Figure 3 jof-10-00773-f003:**
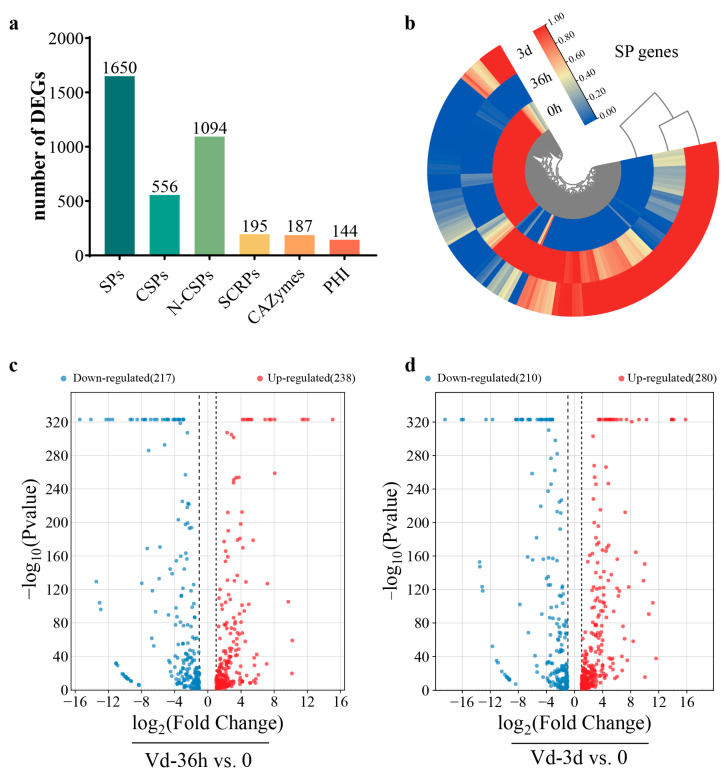
The DEGs encoding secreted proteins (SPs) in *V. dahliae* during infection. (**a**) The number of DEGs encoding SPs in *V. dahliae* during infection. SPs, secreted proteins; CSPs, classically secreted proteins; N-CSPs, non-classically secreted proteins; SCRPs, small cysteine-rich proteins; CAZymes, carbohydrate-activating enzymes; PHI, pathogen–host interaction-associated proteins. (**b**) Circular heatmap of all DEGs encoding secreted proteins during infection. (**c**) Volcano plot of DEGs encoding CSPs in Vd-36h vs. 0. (**d**) Volcano plot of DEGs encoding CSPs in Vd-3d vs. 0. The horizontal axis represents log2 (fold change), and the vertical axis represents −log10 (*p* value). Two vertical dashed lines indicate a 2-fold difference threshold, red dots indicate up-regulated DEGs, and blue dots indicate down-regulated DEGs.

**Figure 4 jof-10-00773-f004:**
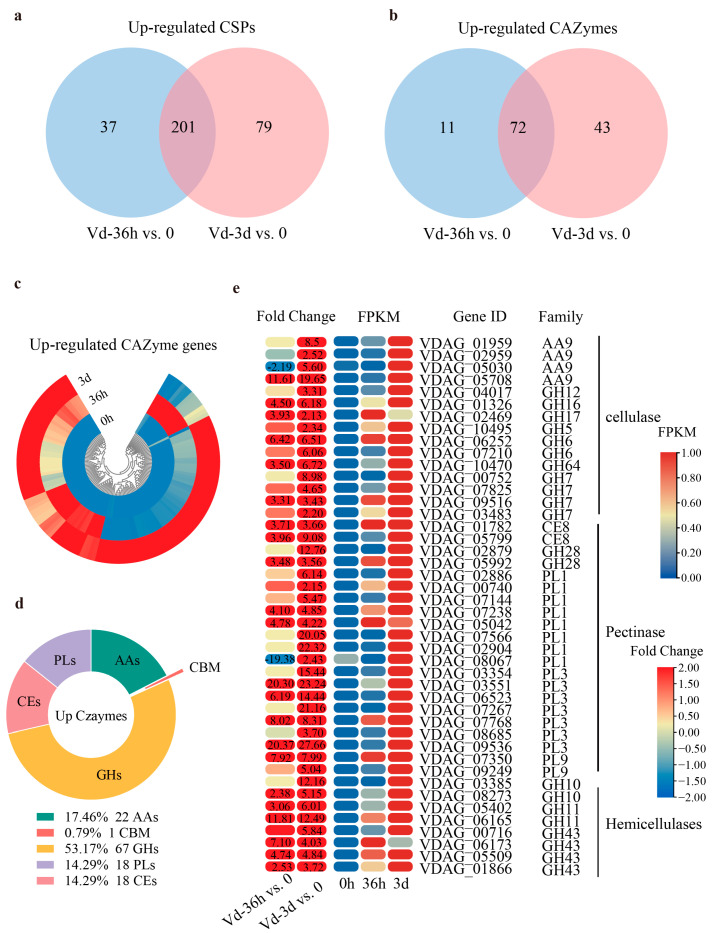
The up-regulated DEGs encoding CAZymes in *V. dahliae* during infection. (**a**) Venn diagram of up-regulated DEGs encoding CSPs in Vd-36h vs. 0 and Vd-3d vs. 0 comparisons. (**b**) Venn diagram of up-regulated DEGs encoding CAZymes in Vd-36h vs. 0 and Vd-3d vs. 0 comparisons. (**c**) Circular heatmap of 126 up-regulated DEGs encoding CAZymes during infection. The heatmap was generated based on the FPKM values. (**d**) Catalog of up-regulated DEGs encoding CAZymes. GHs, glycoside hydrolases; PL, polysaccharide lyase; AAs, auxiliary activities; CEs, carbohydrate esterases. (**e**) Heatmap of up-regulated DEGs encoding CWDEs (cell wall-degrading enzymes) during infection. The heatmap was generated based on the fold change and FPKM values. The numbers in the heatmap are the fold change, with those greater than or equal to 2 displayed. Negative values indicate the down-regulation of DEGs, and positive values indicate the up-regulation of DEGs.

**Figure 5 jof-10-00773-f005:**
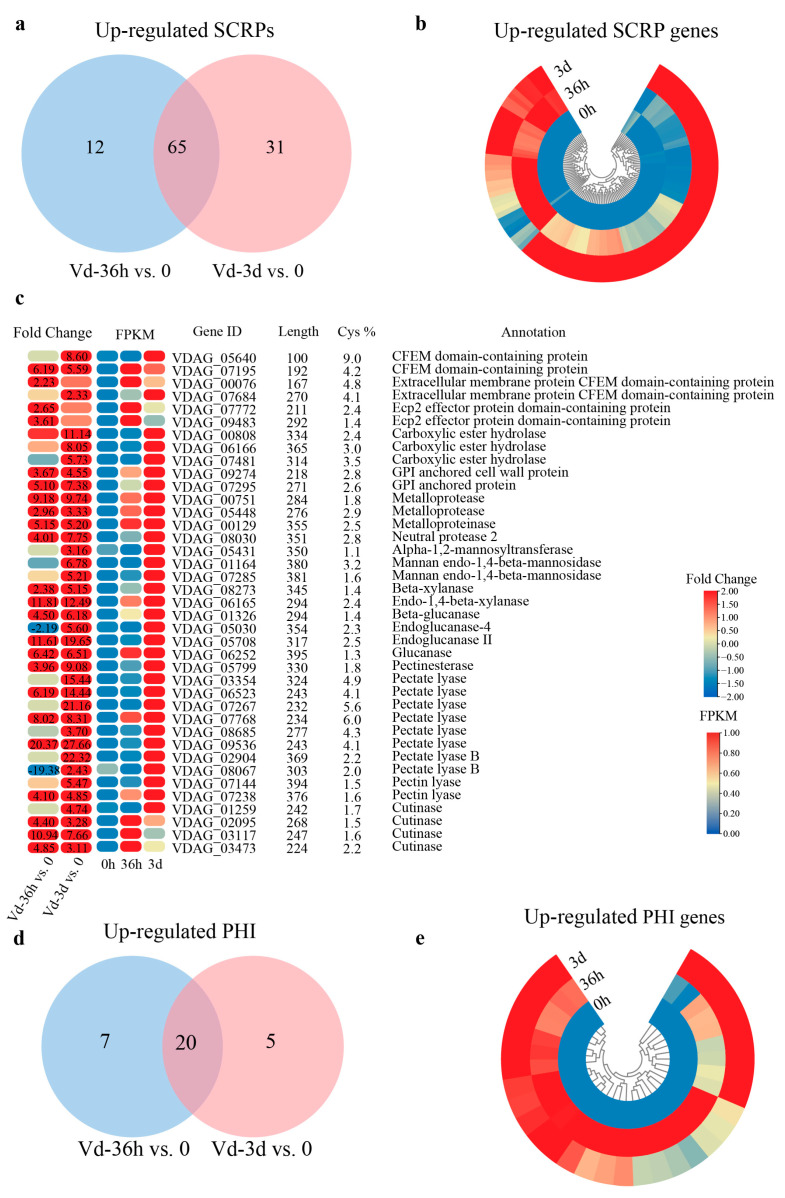
The up-regulated SCRP and PHI DEGs in *V. dahliae* during infection. (**a**) Venn diagram of up-regulated DEGs encoding SCRPs in Vd36h vs. 0 and Vd-3d vs. 0 comparisons. (**b**) Circular heatmap of 108 up-regulated DEGs encoding SCRPs during infection. The heatmap was generated based on the FPKM values. (**c**) Heatmap of partial up-regulated DEGs encoding SCRPs during infection. The heatmap was generated based on the fold change and FPKM values. Numbers in heatmap are the fold change, with those greater than or equal to 2 displayed. Negative values indicate down-regulation of DEGs, and positive values indicate up-regulation of DEGs. (**d**) Venn diagram of up-regulated PHI DEGs in Vd-36h vs. 0 and Vd-3d vs. 0 comparisons. (**e**) Circular heatmap of up-regulated PHI DEGs during infection. The heatmap was generated based on the FPKM values.

**Figure 6 jof-10-00773-f006:**
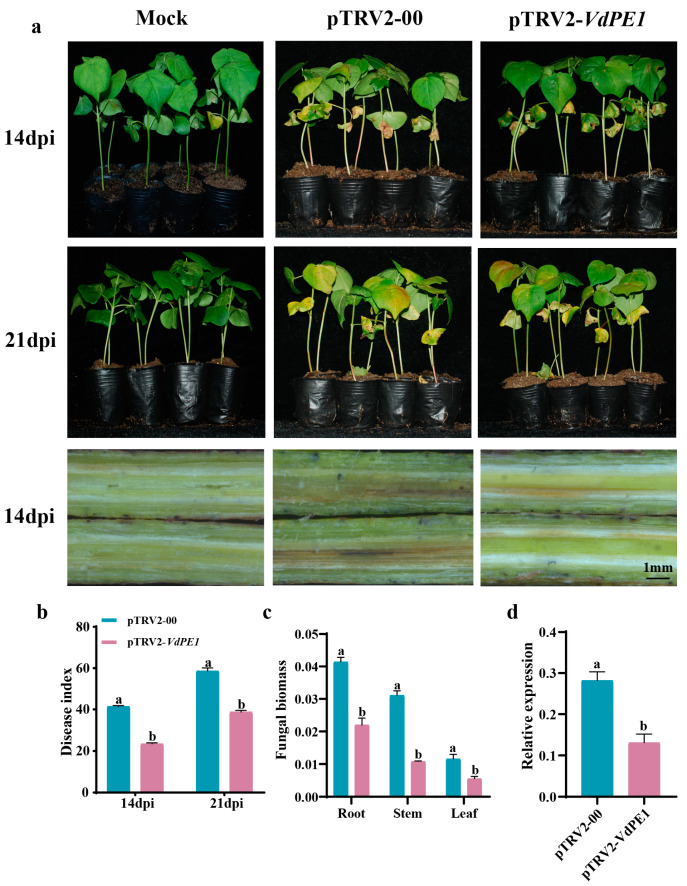
Functional validation of VdPE1 gene in the pathogenicity of *V. dahliae* by HIGS. (**a**) Fungal infection symptoms of HIGS-treated cotton plants at 14 and 21 dpi (days post infection). Vascular browning in the stem segments of HIGS-treated cotton plants was observed at 14 dpi. (**b**) Disease index of HIGS-treated cotton plants at 14 and 21 dpi. (**c**) The qRT-PCR detection of fungal biomass in HIGS-treated cotton plants at 21 dpi. (**d**) The qRT-PCR detection of the expression level of VdPE1 in HIGS-treated cotton roots at 14 dpi. Data were statistically analyzed using IBM SPSS Statistics 26.0. Significant differences between treatments were analyzed by one-way ANOVA using Student–Newman–Keuls (SNK) test. Different letters on bars indicate significant differences at *p* < 0.05.

**Figure 7 jof-10-00773-f007:**
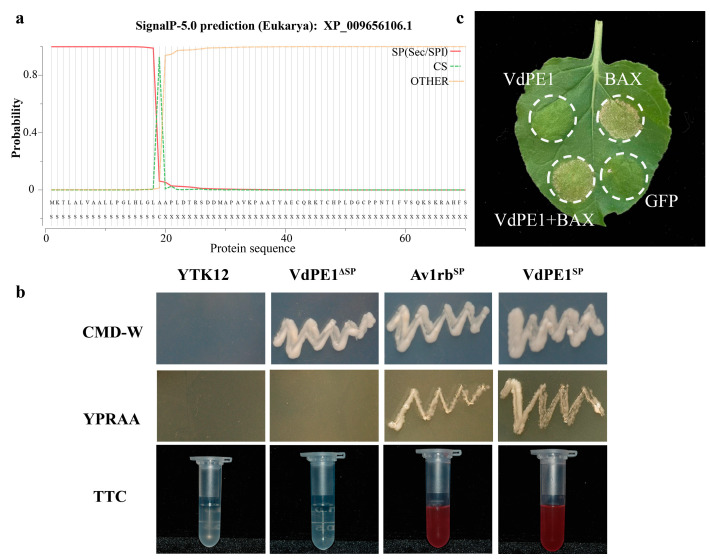
VdPE1 possesses secretory property but fails to induce cell death. (**a**) Predicted signal peptide site of the VdPE1 protein. (**b**) Functional validation of the VdPE1 signal peptide through a yeast signal capture system. The VdPE1 signal peptide sequence was fused in-frame to the invertase sequence in the pSUC2 vector and subsequently transformed into the yeast strain YTK12. Avr1b with known function and VdPE1^ΔSP^ (VdPE1 without signal peptide sequence) were utilized as positive and negative controls, respectively. (**c**) VdPE1 failed to induce cell death in tobacco leaves. Agrobacterium tumefaciens carrying pYBA-1132: VdPE1 was infiltrated into three-week-old tobacco leaves for transient expression. The pYBA-1132: GFP and PYBA-1132: BAX plasmids were employed as negative and positive controls, respectively. Photographs were taken 7 d after infiltration.

**Figure 8 jof-10-00773-f008:**
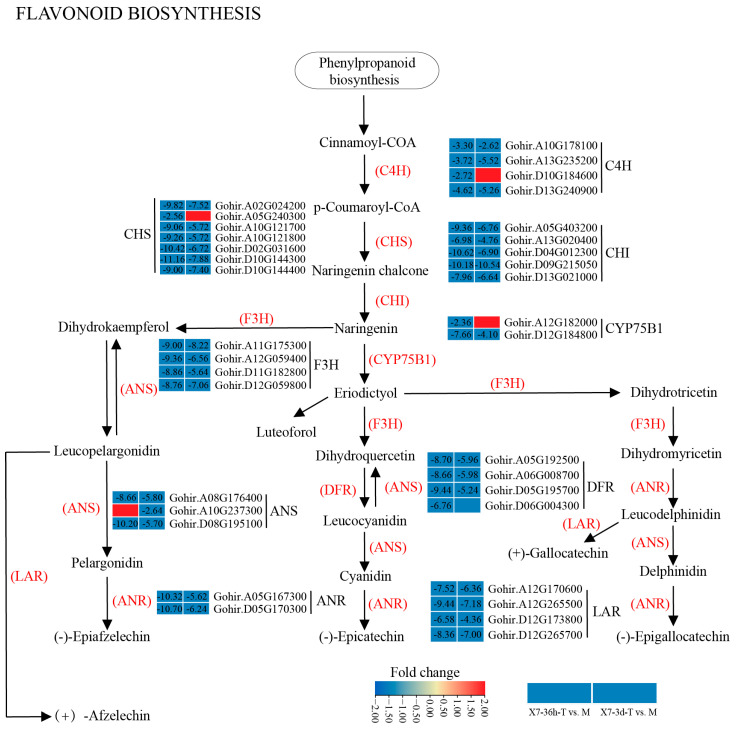
Heatmaps of DEGs involved in flavonoid biosynthesis pathway in cotton. Numbers in heatmap are fold change, with those greater than or equal to 2 displayed. Negative values indicate down-regulation of DEGs. Simple diagram showing relationship of DEGs and flavonoid pathway. C4H, 4 cinnamate 4-hydroxylase; CHS, chalcone synthase; CHI, chalcone isomerase; F3H, naringenin 3-dioxygenase; CYP75B1, flavonoid-3-plus monooxygenases; ANS, anthocyanin synthases; ANRs, anthocyanin reductases; DFRs, dihydroflavonol-4-reductases; LAR, colorless anthocyanin reductases. In X7-36h-T vs. M and X7-3d-T vs. M comparisons, ‘T’ represents infected cotton and ‘M’ represents Mock (water treatment).

**Figure 9 jof-10-00773-f009:**
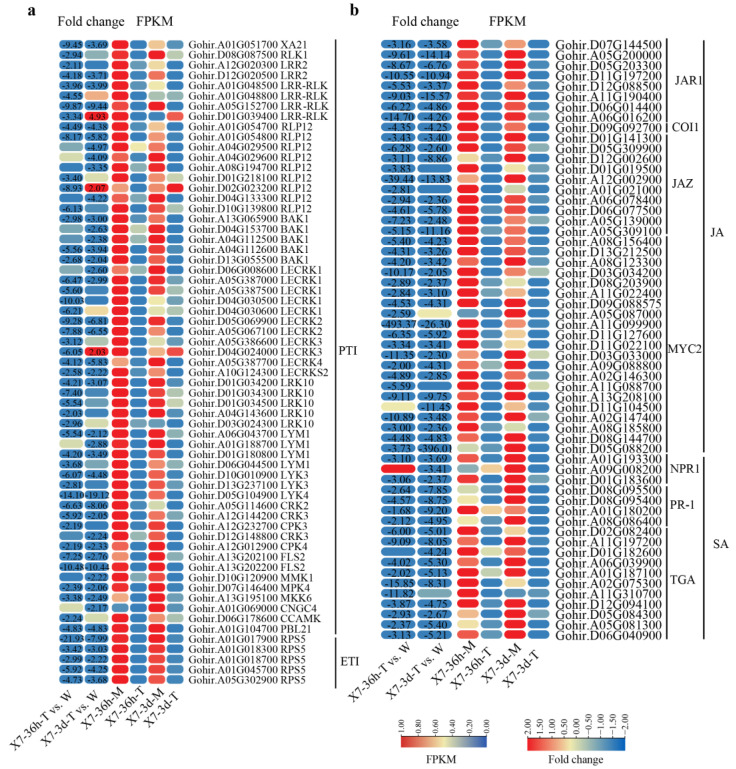
Heatmaps of DEGs involved in PTI and ETI processes and plant hormone signal transduction pathway. (**a**) Heatmap of DEGs involved in PTI and ETI processes. (**b**) Heatmap of DEGs involved in plant hormone signaling transduction pathway. Heatmaps were generated based on fold change and FPKM values. Numbers in heatmaps are fold change, with those greater than or equal to 2 displayed. Negative values indicate down-regulation of DEGs, and positive values indicate up-regulation of DEGs. PTI, pathogen-associated molecular pattern-triggered immunity; ETI, effector-triggered immunity; JA, jasmonic acid; SA, salicylic acid.

**Table 1 jof-10-00773-t001:** The 32 up-regulated PHI DEGs in *V. dahliae* during infection.

Gene ID	Phenotype of Mutant in Other Pathogens	Pathogen Species	Reference
VDAG_00395	Reduced virulence	*Magnaporthe oryzae*	[45]
VDAG_00974	Reduced virulence	*Colletotrichum higginsianum*	[46]
VDAG_01121	Reduced virulence	*Beauveria bassiana*	[47]
VDAG_01774	Loss of pathogenicity	*Parastagonospora nodorum*	No data
VDAG_01922	Reduced virulence	*Magnaporthe oryzae*	[48]
VDAG_01995	Reduced virulence	*Verticillium dahliae*	[49]
VDAG_02250	Reduced virulence	*Verticillium dahliae*	[50]
VDAG_02630	Reduced virulence	*Verticillium dahliae*	[51]
VDAG_02670	Reduced virulence	*Fusarium graminearum*	[52]
VDAG_03620	Reduced virulence	*Verticillium dahliae*	[53]
VDAG_03678	Reduced virulence	*Fusarium graminearum*	[54]
VDAG_03776	Loss of pathogenicity	*Fusarium graminearum*	[55]
VDAG_04213	Reduced virulence	*Fusarium graminearum*	[56]
VDAG_05703	Reduced virulence	*Colletotrichum gloeosporioides*	[57]
VDAG_05890	Loss of pathogenicity	*Verticillium dahliae*	[58]
VDAG_05992	Reduced virulence	*Fusarium oxysporum*	[59]
VDAG_06615	Loss of pathogenicity	*Magnaporthe oryzae*	[60]
VDAG_06763	Reduced virulence	*Verticillium dahliae*	[61]
VDAG_07223	Reduced virulence	*Verticillium dahliae*	[62]
VDAG_07566	Reduced virulence	*Clostridium coccoides*	[63]
VDAG_07697	Reduced virulence	*Verticillium dahliae*	[64]
VDAG_08333	Reduced virulence	*Verticillium dahliae*	[65]
VDAG_09023	Reduced virulence	*Magnaporthe oryzae*	[66]
VDAG_09037	Reduced virulence	*Fusarium graminearum*	[67]
VDAG_09467	Reduced virulence	*Magnaporthe oryzae*	[68]
VDAG_09671	Reduced virulence	*Aspergillus fumigatus*	[69]
VDAG_09689	Reduced virulence	*Verticillium dahliae*	[70]
VDAG_09736	Reduced virulence	*Verticillium dahliae*	[71]
VDAG_09950	Reduced virulence	*Verticillium dahliae*	[71]
VDAG_10099	Reduced virulence	*Fusarium graminearum*	[72]
VDAG_10176	Reduced virulence	*Fusarium graminearum*	[73]
VDAG_10392	Reduced virulence	*Verticillium dahliae*	[74]

## Data Availability

The datasets presented in this study can be found in online repositories. The names of the repository/repositories and accession number(s) can be found at https://www.ncbi.nlm.nih.gov/geo/, accessed on 1 August 2024, PRJNA1108053.

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
