# Peer review of "Dual Transcriptome Analysis Reveals the Changes in Gene Expression in Both Cotton and Verticillium dahliae During the Infection Process"

_jof, 2024, doi:10.3390/jof10110773_

Round 1
Reviewer 1 Report
The approach presented in this study, using dual transcriptomic analysis to address the interaction between plant and pathogenic microorganisms, represents an adequate tool for the analysis of interaction phenomena between organisms. Similar approaches exist in the literature and the information obtained by the computational methodology used in their work also leads to conclusions about the genes up or down-regulated during infection. It is a very efficient and promising tool for subsequent analyses during plant-microorganism interactions since these interactions differ according to the type of plant and the microorganism that attacks it. The work presents the correct follow-up of the protocol and the correct interpretation of the results.
Figures are well presented and explained in figure captions. There is no confusion in them and they are well described. Regarding the references, the authors are local and do not include enough information from international authors. It is important to include more authors from other countries, which means being informed universally. I do not find any faults in the writing of the English language and what was marked are small observations.

Author Response
Comment 1:Figures are well presented and explained in figure captions. There is no confusion in them and they are well described. Regarding the references, the authors are local and do not include enough information from international authors. It is important to include more authors from other countries, which means being informed universally. I do not find any faults in the writing of the English language and what was marked are small observations.
Our response:Many thanks to the reviewer for the affirmation of our work. Adding international author literature is indeed a good suggestion. We have added some literature from international authors. Please refer to references [3],[4],[6] and [22]. Due to the extensive research by local authors, their studies still accounts for a significant proportion in the reference list.
Reviewer 2 Report
The article by the respected authors is a high-tech, large-scale study focusing on interactions in the Verticillium dahliae - Cotton pathosystem. Dual RNA-seq analysis is a new method of transcriptome analysis that allows for simultaneous analysis of both host and pathogen transcriptome profiles, providing a new level of understanding of host-pathogen interactions. The study identified 317 differentially expressed genes (DEGs) encoding classical secreted proteins in V. dahliae, while many resistance-related DEGs were downregulated in infected cotton roots. In addition, the authors included a functional validation of the cell wall-degrading enzyme gene VdPE1. This is a large, relevant study that I think will be of interest to the scientific community. The article is very well illustrated, the materials and methods are described in detail, and the title reflects the content of the article. There are also no remarks on the introduction.
However, I think the article lacks a "conclusion". It is precisely because this study is so extensive that I believe that for the sake of better understanding of the article, the respected authors should add a few paragraphs summarizing the results.
I believe that the article can be accepted for publication after making some minor additions.
The article by the respected authors is a high-tech, large-scale study focusing on interactions in the Verticillium dahliae - Cotton pathosystem. Dual RNA-seq analysis is a new method of transcriptome analysis that allows for simultaneous analysis of both host and pathogen transcriptome profiles, providing a new level of understanding of host-pathogen interactions. The study identified 317 differentially expressed genes (DEGs) encoding classical secreted proteins in V. dahliae, while many resistance-related DEGs were downregulated in infected cotton roots. In addition, the authors included a functional validation of the cell wall-degrading enzyme gene VdPE1. This is a large, relevant study that I think will be of interest to the scientific community. The article is very well illustrated, the materials and methods are described in detail, and the title reflects the content of the article. There are also no remarks on the introduction.
However, I think the article lacks a "conclusion". It is precisely because this study is so extensive that I believe that for the sake of better understanding of the article, the respected authors should add a few paragraphs summarizing the results.
I believe that the article can be accepted for publication after making some minor additions.
Author Response
Comment 1: The article by the respected authors is a high-tech, large-scale study focusing on interactions in the Verticillium dahliae - Cotton pathosystem. Dual RNA-seq analysis is a new method of transcriptome analysis that allows for simultaneous analysis of both host and pathogen transcriptome profiles, providing a new level of understanding of host-pathogen interactions. The study identified 317 differentially expressed genes (DEGs) encoding classical secreted proteins in V. dahliae, while many resistance-related DEGs were downregulated in infected cotton roots. In addition, the authors included a functional validation of the cell wall-degrading enzyme gene VdPE1. This is a large, relevant study that I think will be of interest to the scientific community. The article is very well illustrated, the materials and methods are described in detail, and the title reflects the content of the article. There are also no remarks on the introduction.
However, I think the article lacks a "conclusion". It is precisely because this study is so extensive that I believe that for the sake of better understanding of the article, the respected authors should add a few paragraphs summarizing the results.
I believe that the article can be accepted for publication after making some minor additions.
Our response:Many thanks to the reviewer for this comment. We have already added a clear "Conclusions" section at the end of the discussion part of the manuscript. Please refer to Line 580 to 592.
Reviewer 3 Report
As a reviewer, I could not observe major issues in this work. By the contrary, the work presented is well described. And the manuscript is suitable for its publication
Table 1 I could not find its mention in the main text.
Author Response
Comment 1: Table 1 I could not find its mention in the main text.
Our response:Many thanks to the reviewer for this question. We apologize for this oversight. We have added the mention of Table 1 in the main text. Please refer to Line 380 of the revised manuscript.
Reviewer 4 Report
Based on the results of dual RNA-seq of cotton and Verticillium dahliae during the infection process, the authors found several dozens of genes upregulated in V. dahliae that may provide its pathogenicity. Among these genes, the secreted protein VdPE1 is shown to be important for pathogenesis, although it does not cause death in Nicotiana benthamiana. The work is also of value as a source of data for further functional studies.
Please remove "Title" from the title of the manuscript.
One of the interesting results of the paper is the demonstration that VdPE1 of V. dahliae is important for pathogenesis, please add some words about this in the abstract.
Lines 159, 180, 192, 194. Please put Saccharomyces cerevisiae, Agrobacterium tumefaciens in italics. Please check other parts of the manuscript for correct spelling of genus and species names.
Section 2.5. Please briefly describe how pTRV plasmids work to induce gene silencing.
Section 2.6. Please provide an instrument for the detection of colour changes in TTC.
Line 391. Please give the full name Nicotiana benthamiana.
Author Response
Comment 1: Please remove "Title" from the title of the manuscript.
Our response:Many thanks to the reviewer for this question. The 'Title' has been removed from the manuscript title. Please refer to Line 1.
Comment 2: One of the interesting results of the paper is the demonstration that VdPE1 of V. dahliae is important for pathogenesis, please add some words about this in the abstract.
Our response:Many thanks to the reviewer for this question. Following your suggestion, we have added some related content in the abstract. The added content are as follows ‘A pectinesterase gene (VDAG_01782) belonging to CAZyme, designated as VdPE1, was selected for functional validation. VdPE1 silencing by HIGS resulted in reduced disease symptoms and increased resistance of cotton to V. dahliae.’ Please refer to Line 21 to 24.
Comment 3: Lines 159, 180, 192, 194. Please put Saccharomyces cerevisiae, Agrobacterium tumefaciens in italics. Please check other parts of the manuscript for correct spelling of genus and species names.
Our response:Many thanks to the reviewer for this question. Following your suggestion, We have italicized "Agrobacterium tumefaciens" on lines 159 and 192, as well as "Saccharomyces cerevisiae "on lines 180 and 194. Additionally, we have thoroughly reviewed the entire manuscript to ensure that all genus and species names are spelled correctly. Please refer to Line 168, 196, 212 and 214.
Comment 4: Section 2.5. Please briefly describe how pTRV plasmids work to induce gene silencing.
Our response:Many thanks to the reviewer for this question. TRV (Tobacco Rattle Virus) is a kind of RNA virus, which is made of RNA1 and RNA2. pTRV1 and pTRV2 are virus vectors, which carry cDNA fragments corresponding to RNA1 and RNA2, respectively. After A. tumefaciens carrying pTRV1 and pTRV2-VdPE1 are simultaneously injected into cotton cotyledons, RNA1 and RNA2 of TRV can be generated from the two plasmids, respectively. RNA1 and RNA2 recognize each other to form dsRNA (double strand RNA), which is then cleaved into siRNA (small interfering RNA) by dicer-like enzymes. Then the siRNA can bind with RNA induced silencing complex (RISC) and guide RISC to localization to mRNA strands complementary to siRNA, thereby achieving gene silencing effect. The pTRV2-GhCHLI (The gene mutation causes the leaves to lose their green color) plasmid is usually used as a control. When A. tumefaciens carrying pTRV1 and pTRV2-GhCHLI are simultaneously injected into cotton cotyledons, the cotton GhCHLI will be silenced and the cotton leaves will exhibit bleach phenotype, indicating that the TRV system can function normally. We have briefly described how pTRV plasmids work in section 2.5. Please refer to Lines 168 to 176.
Comment 5: Section 2.6. Please provide an instrument for the detection of colour changes in TTC.
Our response:Many thanks to the reviewer for this question. The experiment only requires a water bath, which can help TTC change colour. We only need our eyes to observe the colour changes in TTC, without the need for other instruments to detect them. We have added the details, Please refer to 207-210.
Comment 6: Line 391. Please give the full name Nicotiana benthamiana.
Our response:Many thanks to the reviewer for this question. We have added the full name of Nicotiana benthamiana. Please refer to Line 417.